# Factors Influencing Physical Activity Participation among Midlife Immigrant Women: A Systematic Review

**DOI:** 10.3390/ijerph18115590

**Published:** 2021-05-24

**Authors:** Ping Zou, Zeest Kadri, Jing Shao, Xiyi Wang, Yan Luo, Hui Zhang, Ananya Banerjee

**Affiliations:** 1School of Nursing, Nipissing University, 222 St. Patrick Street, Suite 618, Toronto, ON M5T 1V4, Canada; 2Faculty of Health Sciences, McMaster University, 1280 Main Street, West Hamilton, ON L8S 4K1, Canada; kadriz@mcmaster.ca; 3Faculty of Nursing, Zhejiang University School of Medicine, 866 Yuhangtang Road, Hangzhou 310058, China; shaoj@zju.edu.cn; 4School of Nursing, Shanghai Jiao Tong University, 227 South Chongqing Road, Huangpu, Shanghai 200025, China; wangxiyi4869@shsmu.edu.cn; 5Faculty of Nursing, Health Science Center, Xi’an Jiaotong University, No. 76 Yanta West Road, Xi’an 710061, China; luoyan0904@xjtu.edu.cn; 6Department of Cardiology, Guizhou Provincial People’s Hospital, Guiyang 550002, China; zhanghui88640@163.com; 7School of Population and Global Health, McGill University, 772 Sherbrooke Street West, Montreal, QC H3A 1G1, Canada; ananya.banerjee@mcgill.ca

**Keywords:** physical activity, participation, midlife, immigrant, women, systematic review

## Abstract

Immigrant women are less likely to be physically active and face many barriers to participation in physical activity. This systematic review aims to identify the influencing factors and adaption approaches of physical activity interventions among midlife immigrant women. A systematic literature search was performed using various databases, such as MEDLINE, PsycINFO, and CINAHL, in February 2021. Studies were included if they investigated midlife immigrant women participating in physical activity interventions and were published in an English peer-reviewed journal in or after 2000. Twenty-two papers were included in this review. Guided by the Ecosocial theory, thematic analysis was utilized for data analysis. Among midlife immigrant women, influencing factors associated with physical activity participation included individual factors (a lack of time, current health status, motivation, and a lack of proficiency in various life skills), familial factors (familial support and seasonality), and community factors (social support and neighbourhood environment). The appropriate adaptation of physical activity interventions included adjustments in language, physical activity intensity, physical activity duration, logistical intervention adjustments and other potential technology-based adjustments. The findings can inform community stakeholders, healthcare professionals and researchers to design appropriate physical activity interventions that meet the needs of midlife immigrant women and improve their health outcomes.

## 1. Introduction

The International Organization for Migration Agency defines migrants or immigrants as individuals who have moved across an international border, away from their habitual place of residence, regardless of legal status, length of stay or cause for movement [1]. In 2019, the global number of international migrants reached about 272 million, which is approximated to be 3.5% of the world’s population [2]. Prevailing trends of international migration can dramatically change population demographics and have significant impacts on health [3]. The healthy immigrant effect refers to the deterioration of health status among immigrants after migration. Initially, upon migrating, immigrants may enjoy a health advantage over native populations, but over time, their health status converges with that of the native population and eventually deteriorates. In later life, many immigrant populations experience a lower health status in comparison to non-immigrant individuals [4]. There is a growing body of research associated with understanding the impact of migration on immigrants’ health status, as they are a vulnerable population with unique healthcare needs [5,6]. Immigrant women have unmet healthcare needs and are less healthy than their male counterparts due to various factors that include socio–cultural gender roles, family caregiving responsibilities, language barriers, transportation concerns, financial constraints and limited health literacy. Immigrant women experience additional barriers towards maintaining their health, despite being able to access the same healthcare resources as non-immigrant women [6,7].

Physical activity refers to people moving, acting and performing within culturally specific spaces and contexts influenced by people’s unique interests, emotions, and relationships [8]. Participation refers to the act of engaging in and contributing to the activities, processes and outcomes of an act, which is essential to sustain the effects of health interventions [9]. Physical activity has many beneficial effects on health outcomes, however, research has consistently shown that immigrant women are less likely to be physically active and face more barriers when engaging in physical activity than non-immigrants [10]. A lack of participation in physical activity may contribute to worse health outcomes among immigrant women. For example, lower levels of physical activity are risk factors for various health conditions such as cardiovascular disease, where the risk of morbidity is already high for immigrant women [11]. The lack of participation in physical activity among immigrant women necessitates the implementation of unique and culturally appropriate health interventions to maintain health.

During midlife, women are subject to more comorbidities and illnesses such as cancer, osteoporosis, cardiovascular disease and depression [11]. Although there is no definitive age for midlife, it is generally accepted that the midlife period usually begins around the ages of 35–40 years and ends between the ages of 60–65 years old [12]. The period of midlife usually implicates important life events as well as other physical, biological and social transitions, such as changes in physical appearance, health, parenting, sexuality and working status. The period of midlife is also considered the life stage implicated by the menopausal transition [13]. Participation in physical activity at midlife in particular is associated with better health outcomes and fewer incidences of major chronic diseases, such as diabetes, myocardial infarction and kidney failure later in life. [14]. Given the lack of participation in physical activity and subpar health status among immigrant women, midlife is a time period where physical activity interventions can be implemented to improve health outcomes.

Reviews in the existing literature have examined the influencing factors of physical activity [15,16,17,18,19]. As early as 2000, Sallis et al. synthesized the correlates of physical activity among children and adolescents [19]. Other reviews examined the influencing factors among culturally diverse populations [17,18], such as South Asian immigrants [16]. Babakus and Thompson also summarized the existing knowledge on physical activity among South Asian immigrant women [15]. Although previous research has addressed the needs, perspectives, and correlates of physical activity among immigrants, there is a lack of research on the experiences of immigrant women in their midlife. With the new publications in the last ten years, an updated review is needed to reflect the current knowledge status. Notably, most existing reviews focused on influencing factors, and no review has summarized the information on adaption approaches, which are necessary to promote physical activity participation.

This systematic review aims to identify the influencing factors as well as adaption approaches of physical activity participation among midlife immigrant women. The Ecosocial theory has been utilized to guide the analysis of psychosocial factors influencing participation in physical activity interventions among midlife immigrant women. The Ecosocial theory is a complex theoretical framework that examines how social determinants influence disease distribution and encourages critical thinking for an appropriate social intervention of health and wellness [20]. In this review, psychosocial factors refer to both individual and social elements that improve or prevent midlife immigrant women from participating in physical activity. Based on the Ecosocial theory, the psychosocial factors in this review were grouped into three categories: individual, familial, and community factors. The research questions of this review are: (1) What are the psychosocial factors influencing participation in physical activity interventions among midlife immigrant women? (2) How can physical activity interventions be adapted appropriately for midlife immigrant women? This review will help to identify the health needs of midlife immigrant women, facilitate their participation in physical activity interventions, and ultimately improve women’s health outcomes.

## 2. Methods

The protocol and reporting of the results of this review were based on the PRISMA guidelines [21].

### 2.1. Eligibility Criteria

Studies were included if they: (a) had studied immigrant women; (b) included participants who were in the middle-age life period, from 40 to 64 years old [12], OR in perimenopausal; (c) investigated interventions with components of physical activity or associated influencing factors towards engaging in physical activity interventions; (c) were published in an English peer-reviewed journal; and (d) were published during or after the year 2000, which would allow for timely results associated with physical activity interventions. Studies were excluded if they: (a) had participant inclusion criteria in a cluster of ages (i.e., 18–60); (b) had participants who had only undergone internal migration in their own country; (c) did not discuss physical activity-related influencing factors/interventions; and (d) did not have an accessible electronic text document.

### 2.2. Information Sources

Various health-related, psychological, sociological and educational science databases, including MEDLINE, PsycINFO, CINAHL, AgeLine, AMED, Health and Psychosocial Instruments, Global Health, ERIC, ProQuest, Nursing and Allied Health Database, PsychARTICLES were selected for the literature search.

### 2.3. Search Strategy and Selection of Evidence

Two reviewers (Z.K. and P.Z.) searched the databases and selected articles for inclusion. The databases were systematically searched using a combination of the following keywords: (Immigrant * OR Newcomer * OR Settle * OR Migrant * OR Noncitizen * OR Incomer * OR Incoming * OR Foreign *) AND (Menopaus * OR Perimenopaus * OR Midlife * OR Mid-life OR Middle-age * OR Middle life Or Menstrua *) AND (Trial * OR Experiment * OR Effect * OR Feasib * OR Pilot OR Random * OR education * OR Physical Activit * OR Physical-Activit * OR Self-Manage * OR Community OR Program * OR Participatory *). Both titles and abstracts were used in the field of search. The citations were exported into EndNote to remove any duplicates. The titles and abstracts of all citations were screened for the relevance based on the established eligibility criteria. All eligible articles were searched for full-text documents and the full-text documents were carefully reviewed by two reviewers (Z.K. and P.Z.). The most recent search was conducted in February 2021. A total of 485 records were identified from databases and 22 studies that were included in this review (Figure 1).

### 2.4. Quality Assessment

The Critical Appraisal Skills Program Checklists (https://casp-uk.net/casp-tools-checklists/ (accessed on 5 March 2021)) was used as a quality assessment tool to assess included articles [22]. Since CASP has distinct appraisal checklists designed for various research designs, we used specific scales according to the type of studies included. These checklists are not designed to generate a final quantitative score, rather, they draw attention to elements of a rigorous study and evaluate a study as a whole. Using these checklists, we were able to classify the quality of included papers as low, moderate or high. Two reviewers (Z.K. and P.Z.) independently evaluated each article; any discrepancies in ratings were discussed along with the guidelines until consensus was reached. No paper was excluded due to low quality. Thus, all papers included in this review were at moderate-to-high quality.

### 2.5. Risk of Bias in Studies

In order to verify the reliability and the degree of agreement of the coding during the evidence selection and data extraction, two reviewers (Z.K. and P.Z.) were involved in this process. The risk of bias of each eligible article was assessed by adopting a dichotomous nominal scale of two unique values (yes/no), which was developed to assess the concordance in the 22 included papers. The degree of agreement obtained in the classification of the papers was 95%, which was calculated by dividing the number of coincidences by the total number of classification and multiplying it by 100.

### 2.6. Data Extraction

Data were independently extracted by two reviewers (Z.K. and P.Z.) based on pre-determined criteria. From each article, various data, including the authors, year of publication, study population, research design, recruitment method, intervention information, sample size, sample characteristics, comparison group, outcomes, measurements, and significant findings were extracted. The data were collected and organized into an Excel spreadsheet. The reviewers discussed disagreements in data extraction until consensus was reached.

### 2.7. Data Analysis and Synthesis of Results

Once the data were organized inExcel (v16.0, Microsoft Corporation, Washington, USA), descriptive statistics were used to present the characteristics of included studies. Thematic analysis was then used to summarize the findings to each research question [23]. After the thorough extraction of qualitative and quantitative data separately, data were recontextualized in relation to the research questions [24]. The Ecosocial theory allowed us to broadly categorize the data at the individual, familial and community levels [20]. Further categorization involved analyzing similar information to identify relevant sub-themes at the individual, community and social level. Categorization results were compared among reviewers (Z.K. and P.Z.) and any disagreements among reviewers were resolved with a consensus decision.

## 3. Results

### 3.1. Characteristics of Included Studies

Twenty-two papers were included in the review (Table 1). Two (10%, 2/22) studies were published between 2000 and 2010, seven (32%, 7/22) between 2011 to 2015, and thirteen (59%, 13/22) between 2016 to 2021. Twelve (55%, 12/22) studies were conducted in the United States, eight (36%, 8/22) in South Korea, one (5%, 1/22) in Canada, and 1 (5%, 1/22) in New Zealand. Sample sizes ranged from 16 to 222 and 12 (55%, 12/22) studies had no more than 50 participants. Among the included studies, 12 (55%, 12/22) were quantitative in design, 6 (27%, 6/22) were qualitative, and 4 (18%, 4/22) involved mixed methods. Of the 12 quantitative studies, seven (58%, 7/12) were randomized trials, three (25%, 3/12) were quasi-experimental designs, and two (17%, 2/12) were pre- and post-tests.

### 3.2. Psychosocial Factors

#### 3.2.1. Individual Factors

##### Lack of Time

Seven (32%, 7/22) studies addressed a lack of time as a barrier influencing attendance, participation and/or adherence in physical activity interventions [25,26,27,28,29,30,31]. A lack of time was mainly due to household responsibilities and work obligations [25,26,27,28,29,30]. A study among Korean–Chinese immigrant women identified that individual occupational related time constraints were related to unstable employment, unfamiliar working conditions, unpredictable hours at work, lack of rest time, frequent changes of workplaces, difficulty in finding new work, long working hours, and/or increased job demands [30]. A study among Korean–Chinese migrant women found that a lack of time due to occupational-related duties left them feeling too busy, tired or stressed [31].

##### Health Status

Four (18%, 4/22) studies addressed the individual health status as an influencing factor towards physical activity participation [28,30,31,32]. Optimal physiological and mental health conditions could facilitate participation in physical activity. Midlife South Asian immigrant women identified that optimal physical and psychological health, in addition to the emphasis on the external beauty of participants through interventions, was able to promote positive attitudes towards physical activity [28]. Subpar health status prevented participation in physical activity interventions. A stretching intervention among middle-aged Korean–Chinese women found that illness and fatigue due to participants’ current health status affected intervention adherence [30]. Another study among Korean–American women found that the experience of pain, sickness or discomfort, prevented women from engaging in physical activity [31]. Suboptimal health status was a facilitator for physical activity. A study among middle-aged Latinas participating in a health intervention found that professional health assessments motivated them to engage in more leisure-time physical activity, as a suboptimal health status would make them more conscious of their responsibility to take care of their health [32].

##### Motivation

Four (18%, 4/22) studies identified that motivation was also an influencing factor associated with participation in physical activity interventions [28,30,31,32]. Identified reasons for a lack of motivation among midlife immigrant participants were fatigue, lack of confidence and lack of interest [28,30]. Other studies suggested that motivation could be improved through facilitating professional health assessments, learning through skits and role play, attaining optimal health status among participants, as well as gaining confidence through small achievements [31,32].

##### Lack of Proficiency in Life Skills

One (5%, 1/22) study found that a lack of proficiency in life skills such as language skills, driving ability, and physical activity-related knowledge were barriers associated with participation in physical activity interventions [27]. A study on the development of culturally relevant walking interventions among midlife Korean–Chinese women identified language barriers, a lack of knowledge of the benefits of physical activity and recommended exercise intensity as barriers influencing engagement. These women suggested that such barriers could potentially be overcome through the use of an English-speaking facilitator who would consider the language skills of participants when addressing their needs [27].

#### 3.2.2. Familial Factors

##### Familial Support

Five (23%, 5/22) studies reported that family support was an influencing factor associated with physical activity participation [28,30,31,32,33]. Family support was an important facilitator. A promotora-led intervention among midlife immigrant Latinas identified that family member encouragement facilitated intervention participation [32]. Another study among Arab–Canadian women identified persuasion and support by children as a contributor to intervention motivation and engagement [33]. Additionally, the presence of a supporting family was a facilitator for physical activity among Korean–Chinese migrant women. A woman’s spouse might serve as a way to support physical activity participation by checking their progress, reminding them of their goals, setting exercise equipment at home and also exercising together [31]. A lack of family support was a barrier associated with physical activity intervention participation. Negative comments from an individual’s husband may decrease their desire to participate in physical activity [31]. Identified literature among South Asian and Korean–Chinese migrant women identified that a lack of family member support was associated with decreased engagement in physical activity [28,30].

##### Seasonality

Three (14%, 3/22) studies identified seasonality as a familial barrier associated with engaging in physical activity interventions [30,34,35]. Seasonality refers to certain times of the year (dependant on the host country’s holiday season/calendar) that immigrant women utilized to fulfil family-related endeavors. A health-education intervention study conducted among midlife South Asian American women found that continuity and follow-up was affected by seasonality during times of the year when children were out of school (i.e., winter break, summer break) or when participants were using this time to visit their native South Asian countries [35]. In addition, seasonality was also a familial influencing factor, as participants used these times (i.e., Christmas) to visit family in their native countries [34]. A study which investigated barriers to stretching among midlife Korean–Chinese migrant woman identified that during the holiday season, their families in China would visit them in Korea, which would result in extra work during certain seasons for these women, leaving less time for personal matters [30].

##### Familial Obligations

Three (14%, 3/22) studies addressed familial obligations as barriers affecting engagement and adherence to physical activity interventions [28,30,33]. These included role expectations of South Asian women to prioritize their family’s needs before their own as wives, mothers and/or daughters-in-law, which reduced the potential time they had to engage in physical activity [28]. Among Korean–Chinese midlife women, commitments to family and relatives was a significant barrier to performing stretching exercises, due to responsibilities such as assisting family and relatives with moving, visiting them in hospitals or helping them settle in Korea [30]. Additionally, women identified that childcare was an important consideration associated with participating in the program as they prioritized family obligations over personal health improvements [33].

#### 3.2.3. Community Factors

##### Social Support

Five (23%, 5/22) studies addressed social support as an influencing factor towards engaging in physical activity [28,30,31,32,33]. The presence of social support was found to be a facilitator associated with engagement in physical activity interventions. For example, community interactions with Latina promotoras throughout the intervention allowed for participants to have increased motivation in the form of increased physical and emotional guidance, support and knowledge [32]. Community motivational groups were identified to be facilitators for engagement in physical activity among South Asian women [28]. Another study identified the facilitator of social support for physical activity among Korean–Chinese migrant women. They stated that having an exercise buddy for walking as well as having a sense of belonging facilitated their engagement in the intervention [31]. On the other hand, studies also identified that a lack of social support was a barrier associated with engagement in physical activity interventions in various cohorts of midlife immigrant women [30,33].

##### Neighbourhood Environment

One (5%, 1/22) study demonstrated other community-related aspects which influenced engagement and participation in physical activity [31]. Korean–Chinese migrant women demonstrated that feeling unsafe was a barrier to engaging in physical activity, whilst a neighbourhood that was desirable for physical activity was identified as a facilitator [31].

#### 3.2.4. Intervention Adaptation

##### Language Accommodation

Nine (41%, 9/22) studies addressed language accommodations specific to the needs of midlife immigrant women participants [27,28,32,36,37,38,39,40,41]. A bilingual research team or physical activity instructor who spoke English and the respective language of the participant cohort facilitated language accommodation [28,32,36,39,40,41]. Another way that language accommodations were made was by making intervention materials available in the respective languages of participants. A lifestyle intervention among Latinas utilized informational brochures available in Spanish that were accessible and easy to read [32]. Similarly, intervention materials for perimenopausal Latinas made intervention materials available in English and Spanish [40]. A web-based intervention for Korean–Chinese midlife women included module and program functionalities which were available in English, Mandarin Chinese and Korean [37]. Likewise, another web-based physical activity intervention conducted among midlife Asian–American women integrated web components which were available in English, Mandarin Chinese and Korean [38]. Moreover, a study investigating the development of a culturally relevant walking program among Korean–Chinese women considered cultural differences and the level of Korean language comprehension of participants when designing their intervention [27].

##### Adjustment of Physical Activity Intensity/Duration

Five (23%, 5/22) studies made adjustments in terms of exercise intensity and duration to meet the individual needs and experiences of participants [25,30,34,37,42]. A study conducted on midlife Taiwanese immigrants in New Zealand utilized stretching exercises that were designed by the New Zealand Accident Compensation Corporation Sportsmart organization to adjust exercises based on the needs and experiences of the participant cohort [34]. One intervention among Korean–Chinese immigrant women selected only 1–3 movements for each body region in order to decrease participant burden and designed a 6 min program with stretching exercises, which could be done conveniently at any time or place [30,42]. In a dance intervention, choreographies were specifically designed so participants could conveniently adjust them to their own comfortability and experience [25]. In addition, a web-based intervention for Asian–American midlife women designed culture-specific exercises that could easily be completed with families without necessary equipment [37].

##### Logistical Adjustment

Five (23%, 5/22) studies incorporated various logistical adjustments by implementing interventions at convenient locations and appropriate times with features such as childcare services and transportation vouchers [27,33,39,40,43]. Most of the included studies that were conducted in-person took place in convenient settings such as community centers, religious centers and even individual participant homes which were agreed upon by researchers and participants. In particular, one study was held in a setting within a 5-mile radius of the participants’ residence [39]. Another study conducted among midlife Korean–Chinese women found that both location and time were convenient for participants when orientations and assessments were performed at a migrant resource center on Sundays [27]. These findings regarding setting convenience were in contrast to the findings of an mHealth physical activity intervention among Korean–Chinese immigrant women, which suggested that home-based or mobile interventions were more feasible for a cohort of midlife immigrant women in place of on-site interventions, due to convenience [43]. Complimentary services and incentives were also provided to participants throughout the intervention via transportation vouchers and childcare services [33,40]. One study offered childcare services during the intervention sessions for participants with children and offered taxi vouchers for participants with no means of transportation [33]. A physical activity intervention among perimenopausal Latina women provided financial incentives for baseline and subsequent 6-month and 12-month visits, where both transportation vouchers and childcare services were offered to participants [40].

##### Use of Mobile-Based Technology

Five (23%, 5/22) studies utilized mobile-based technologies for various purposes including intervention administration, and motivational or assistive reminders [40,42,43,44,45]. The administration of a walking intervention among Korean–Chinese midlife women found that the use of a mobile-phone-based pedometer was practical for accurately self-monitoring daily physical activity [44]. Another mHealth intervention among Korean–Chinese midlife immigrant women found that administering an intervention through phones was cost-effective since over 91% of individuals in the study region utilized smartphones [43]. In addition, smartphone technology was utilized for the purposes of information sharing, motivation, or assistance with engaging in physical activity [40,42,43,44,45]. For example, a pedometer-based walking intervention utilized 12 exercise-related motivational text messages with enhanced images of medals to promote adherence and improve acculturation [45]. A mobile-based health intervention development strategy among Korean–Chinese midlife women utilized assistive reminders through text-messaging to encourage and provide emotional support for the continuation of the program [43]. A physical activity intervention among Latina women provided assistance to participants by reviewing missed class content over the phone and providing motivational reminders for the next class [40].

## 4. Discussion

### 4.1. Summary of Findings

Among midlife immigrant women, a wide range of influencing factors were associated with participation in physical activity interventions. Identified influencing factors at the individual level were a lack of time, current health status, motivation, as well as a lack of proficiency in various life skills. Familial factors included familial support and seasonality. Influencing factors at the community level were social support and neighbourhood environment. To meet the needs of midlife immigrant women, the appropriate adaptation of physical activity interventions included adjustments in language, physical activity intensity, physical activity duration, logistical intervention adjustments and other potential technology-based adjustments.

### 4.2. Individual Factors

Findings of our review suggested that a lack of time observed by midlife immigrant women due to occupational/work responsibilities, household responsibilities and other individual responsibilities affects participation and adherence in physical activity interventions. These findings are in accordance with another study involving mixed ethnicities of midlife women who identified that a lack of time is a common barrier to participating in physical activity due to their multiple role-taking responsibilities as women [46]. It was found that this is barrier is common not only among ethnic minority women, but also white women, due to work, family and household duties [46]. Other studies also show consistent findings; it has been observed that a lack of time due to work responsibilities is a barrier associated with preventative breast cancer screening among populations of immigrant women in Spain. This is due to the fact that these immigrant women had strict job schedules with job security concerns, resulting in a reluctance to take time off work [47]. Due to these time constraints among immigrant women, it is imperative that future interventions consider the work schedules and availability of this target population.

Motivation was found to be an influencing factor for participation in physical activity interventions in our review. Our review found that a lack of motivation was caused by fatigue, lack of confidence and lack of interest, which could be improved through the use of skits, role play and other small achievements [28,30]. These findings are in contrast to another study exploring motivation for physical activity among asylum seekers in Northern England. It was found that motivation was dependent on the type of physical activity, the discipline required to maintain it as well as whether it was personally perceived as enjoyable by participants [48]. Additionally, motivation was difficult for some individuals due to competing responsibilities as asylum seekers [48]. These mixed findings demonstrate that motivation for physical activity is a complex influencing factor, which needs to be evaluated through a case-by-case basis to ensure that midlife immigrant women are able to participate and adhere to physical activity interventions.

Our review found that an inadequate proficiency in various life skills such as language skills, driving skills and physical activity knowledge were barriers affecting participation and adherence in physical activity. Consistent with our findings, a study suggested that limited proficiency in a second language among older female Turkish women migrating to the Netherlands resulted in personal insecurity, which negatively impacted the formation of social relationships, feelings of belonging and wellbeing [49]. In contrast to our review’s findings, a study among migrants found that despite high self-rated English language skills, they still experienced communication difficulties associated with accessing appropriate health services [50]. These findings suggested that simply possessing language skills may not be enough for immigrant populations to access health services. Future physical activity interventions for midlife women should include intervention elements that facilitate the learning of new language skills and establish environments where women feel comfortable communicating in their native languages.

### 4.3. Familial Factors

Findings of our review indicated that familial support was associated with engagement and participation in physical activity interventions. Consistent with our findings, a study has shown that general family member support among Mexican–American women influences levels of physical activity participation through perceived support from their spouse [51]. In non-married individuals, a higher level of non-spousal family support was associated with a lower risk of development of heart problems among unmarried women [52]. Both of these findings are consistent with our study, demonstrating that educational support and family-oriented approaches should be implemented for spouses and other family members of midlife immigrant women to improve participation in physical activity interventions.

Findings of our review indicated that seasonality was an influencing factor for participation in physical activities, as immigrant women use holiday seasons to visit families and may also face unique familial circumstances during these times. These findings are in contrast to a study investigating older Chinese immigrants, which found that due to their personal activities, working status or familial responsibilities, the appropriate times to meet with elder Chinese individuals were on weekends and holidays since they might have breaks from familial responsibilities during this time [53]. Another study’s findings are in accordance with our review, which also demonstrate that women might face challenges during the holidays due to unique familial circumstances. In fact, an intervention aiming to increase participation in cervical cancer screening among immigrant women in Norway had to extend their recruitment period, as it was hindered by two seasonal occasions. The first was during the Easter holidays, as schools were closed for two weeks and many of the women were travelling. The second circumstance was the religious month of Ramadhan and summer vacation [54]. These findings demonstrate that both recruiting and retaining immigrant participants requires accommodation, flexibility as well as awareness of the various life and familial circumstances of potential participants.

### 4.4. Community and Social Influencing Factors

Findings of our review indicated that social support from community members through community networks, exercise buddies, and health coaches/promotoras can affect participation in physical activity among midlife immigrant women [28,32]. Another study investigating friend and co-worker social support with physical activity found that after controlling for covariates, co-worker support was positively associated with participation in physical activity [55]. These findings are consistent with our study, as it demonstrates that specific social networks may be more beneficial towards engagement and participation in physical activity. Future interventions should consider the particular social networks of women and encourage women to participate in physical activity interventions with their existing social networks. In addition, our review found that most discussed factors were focused on the individual rather than the community level. This finding is consistent with other research [56]. The Ecosocial theory allows a broad view of health behaviour causation, with the community, social, and physical environment included as contributors to physical activity. The application of this theory can facilitate the exploration of factors, particularly those outside the health sector, such as child care services and transportation systems [56].

### 4.5. Intervention Adjustment

Findings of our review indicated that language accommodation, such as the incorporation of the participants’ native language in the intervention process, facilitates participation. Other research also found that older women who are less educated and may have other issues, such as spouse dependency/lack of independence, possess lower levels of language proficiency of their host country and have fewer opportunities to improve that language. The lack of language proficiency of a host country serves as a barrier to accessing health care and community services [57]. These findings are in accordance with our review, demonstrating the need for tailored interventions with appropriate language accessibility to accommodate participants with different levels of language proficiency of the host country.

The findings of our review indicated that the adjustment of intensity levels and duration of physical activity is necessary to meet the needs of midlife immigrant women. These findings are in accordance with another study that explored the comfort levels of physical activity among immigrants and refugees in Minnesota, United States (US). It was found that gym/physical activity culture among these populations is unfamiliar compared to US-born individuals due to a lack of knowledge on how to exercise in a gym or use gym equipment [58]. Due to the possible unfamiliarity of physical activity culture and equipment to immigrants, the adjustment of the intensity and duration of physical activity is an important factor associated with adapting interventions.

The findings of our review indicated that logistical adjustments, such as the use of transportation vouchers, promoted participation and adherence among immigrant women [33,40]. These findings are in accordance with another health coaching and transportation assistance intervention, which found that a bus token intervention improved the follow-up of participants by at least 40% [59]. Thus, community-based interventions should be promoted [60,61]. The usage of taxi vouchers or bus passes to overcome transportation concerns should be considered as an implementation strategy.

Findings of our review also indicated the benefits of using smartphone-based technology for the purposes of motivation and assistance [40,42,43,44,45]. Our findings are in accordance with other studies using phone text-message psychosocial intervention therapies, which found significant improvements in mood and depression among vulnerable groups of immigrant women, challenging previous notions that phone-based navigation for immigrants is difficult [62,63]. However, our findings are in contrast with another study which suggests that there may be disparities in technology-use between native older adults and immigrant older adults due to disparities in language proficiency, health literacy, socioeconomic status, education and acculturation levels [64]. These findings demonstrate that technological-based adaptation in an intervention can potentially improve motivation and assistance in physical activity interventions; however, there must be measures in place to ensure immigrants are able to appropriately access and utilize technology-based intervention components.

## 5. Limitations

This review presents a range of influencing factors and adaptations that researchers and practitioners can use to design physical activity interventions for midlife immigrant women. However, it does not identify one singular optimal strategy. In practice, specific considerations are needed to tailor an intervention according to the characteristics of the target group. Furthermore, the range of included studies varied in quality and research design, as studies were comprised of both qualitative and quantitative methods. The included studies not only consisted of already administered physical activity interventions, but also those that investigated factors regarding the development of interventions that had not yet been undertaken. This made it difficult to provide concrete interpretations and to synthesize the identified influencing factors and adjustment components. Finally, a methodological limitation to this review was the very specific inclusion criteria relating to middle age. This meant that studies conducted in a cluster of ages (i.e., 18–50 years) were excluded despite potentially comprising of mostly midlife women. There is a possibility that this may have taken away from the more in-depth information regarding influencing factors associated with participating in physical activity interventions. However, this strict inclusion criterion maintained the integrity of the study, ensuring that findings were specific only to groups of midlife immigrant women.

### Implications and Future Studies

This review provides meaningful references to healthcare professionals, researchers and policymakers in implementing appropriate health interventions among underserved populations of midlife immigrant women. Firstly, the findings of this study can contribute to evidence-based physical activity intervention planning [56]. Our findings can be used as a guide for healthcare professionals and stakeholders in the community to create appropriate and inclusive interventions in the practice of preventative care to meet the physical activity needs of midlife immigrant women. For example, policymakers may be able to establish guidelines that provide midlife immigrant women with appropriate accommodations (i.e., city bus pass, subsidized childcare services) to further incentivize women to participate in physical activity interventions. Secondly, the findings of this study will guide community workers in implementing community-based strategies to increase participation, adherence and engagement in physical activity interventions so that they are inclusive towards the needs of midlife immigrant women. Third, our findings will facilitate policy makers to evaluate, refine, and better incorporate midlife immigrant women’s needs into current physical activity policy [65]. Fourth, while this review systematically summarizes the factors influencing physical activity among heterogenous immigrant populations, further research is needed to explore more specific or unidentified influencing factors among specific immigrant groups that were not addressed in this review. Fifth, future research should continue using the Ecosocial theory to explore the factors influencing physical activity at the community and societal levels. Future theoretical models should strive to explore how local healthcare systems/institutions, health policies, gender norms and social taboos surrounding health influence participation in physical activity [56,66]. Finally, there is a lack of literature on non-physical activity interventions that are specifically available for midlife immigrant women. This necessitates further research in other types of interventions, such as dietary interventions, which can potentially contribute to better physical and mental health outcomes among midlife immigrant women [65]. Future research should address the effectiveness of physical activity in addition to other lifestyle interventions on health outcomes among midlife immigrant women.

## 6. Conclusions

This review summarizes the influencing factors at the individual, familial and community levels in addition to the adaptation strategies of physical activity interventions for midlife immigrant women. Individual factors were a lack of time, current health status, motivation and a lack of proficiency of various life skills. Familial influencing factors were family support and seasonality. Community factors included social support and neighbourhood environment. Additionally, adapting physical activity interventions should consider various factors, such as adjustments in language, physical activity intensity, physical activity duration, logistical intervention adjustments and technology-based adjustments. These findings can inform community stakeholders, healthcare professionals and researchers to design optimal physical activity interventions that meet the needs of midlife immigrant women and improve their health outcomes. Future studies should further explore influencing factors in the community and societal level and the impacts associated with physical activity interventions among specific subpopulations of midlife immigrant women.

## Figures and Tables

**Figure 1 ijerph-18-05590-f001:**
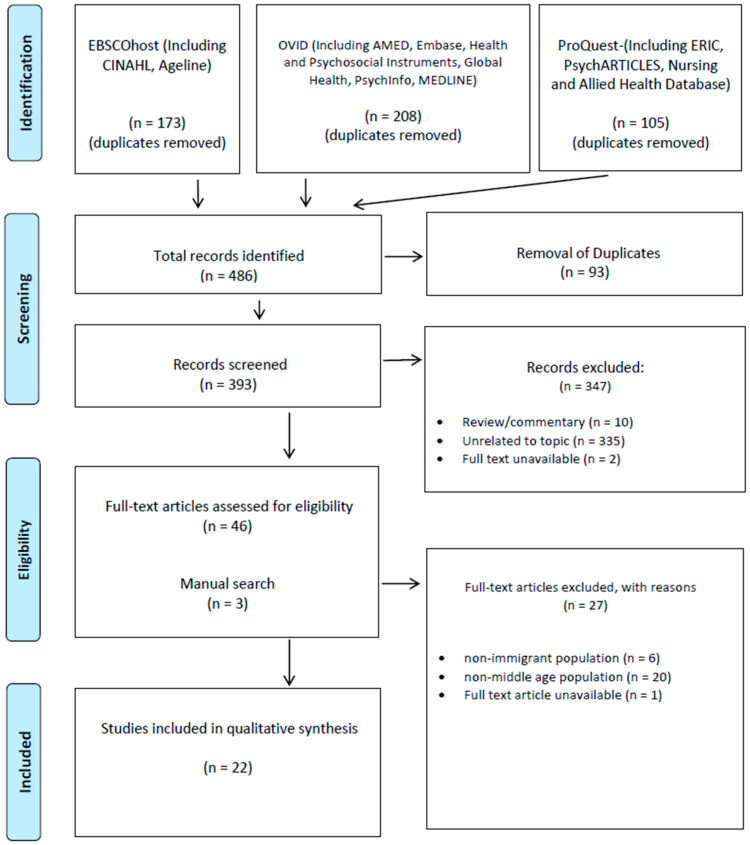
Searching process.

**Table 1 ijerph-18-05590-t001:** Summary of included studies.

Author(s)/Year/Research Setting	Research Design Qualitative/ Quantitative/ Mixed Methods	Research Purpose	Sample Size/Participants/Mean Age (SD)	Findings
Chen and Watson (2002) New Zealand	Quantitative: randomized control trial	To explore the effects of short-term diet and exercise intervention on body fat levels in middle-aged women from Taiwan, who have immigrated to New Zealand.	30 midlife Taiwanese immigrant women living in New Zealand Mean age: 47 (5.3) in intervention group, 45 (4.5) in control group)	No significant differences in body weight change between both groups, body fat levels in intervention group decreased significantly. Participants with higher initial body weight, body mass index, and participants who were more active lost more weight during the intervention.
Shakil (2010) USA	Quantitative: quasi-experimental	To examine awareness of osteoporosis prevention among peri- and postmenopausal South Asian women with a health education intervention.	61 midlife South Asian–American immigrant women Mean age: 52.30 (8.72)	Women who completed the educational intervention program were better prepared to prevent and manage osteoporosis. Post-test found highest knowledge gains on the following: on facts and healthy behaviours for preventing and managing osteoporosis, (i.e., through physical activity).
Lee (2014) South Korea	Quantitative: randomized control trial	To assess the efficacy of a 12-week self-managed, community-based stretching program on musculoskeletal fitness, musculoskeletal symptoms and acculturative stress levels using an enhanced intervention (EI) and standard intervention (SI).	59 middle-aged Korean–Chinese immigrant women Mean age: 53.03 (5.46)	Significant increase in work-related musculoskeletal disorder knowledge from baseline values at 12-week assessment for EI (*p* < 0.001) and SI (*p* = 0.013) group. Significant increases in flexibility from baseline values at 12-week assessment for EI (*p* < 0.001) and SI (*p* < 0.001) groups. Muscle strength and work-related musculoskeletal disorder symptoms: no significant changes at the 12-week follow-up in both groups. EI group: no statistically significant decrease in perception of acculturative stress at the 12-week follow-up; SI group: perceived less acculturative stress at 12 weeks than at baseline (*p* < 0.001).
Koniak Griffin (2015) USA	Quantitative: randomized control trial	To evaluate knowledge of cardiovascular disease (CVD) among overweight middle-aged immigrant Latinas who spoke little-to-no English and participated in an educational intervention.	90 middle-aged Latina/Hispanic–American immigrant womenMean age: 42.60 (7.0)	Baseline knowledge that heart disease is the leading cause of death in women was low. Comparison of preintervention and postintervention scores on questionnaire showed significant changes in knowledge about cardiovascular disease (7.9 (SD, 2.6) and 9.4 (SD, 1.0)). Scores for individual items correctly answered significantly improved for 9 of the 11 items, including questions about portion control to lose weight, physical activity, overweight, and risk for heart disease. Knowledge of heart-healthy diet behaviour increased, as well as availability of early treatment after the onset of heart attack or stroke symptoms.
Koniak Griffin (2015) USA	Quantitative: randomized control trial	To evaluate the outcomes and feasibility of a promotora-led lifestyle behaviour intervention for overweight, immigrant Latinas.	223 immigrant Latina/Hispanic–American immigrant women Mean age: 43.30 (7.44)	Overall dietary scores improved for women who participated in the intervention. Significant decrease in physical activity behaviours among the control group, whereas the intervention group remained persistent with their physical activity habits. Observed improvement in pre- and post-test scores. No significant difference in retention rates for intervention between both groups.
Chee et al. (2016) USA	Quantitative: randomized control trial	To examine the efficacy of web-based physical activity promotion program towards enhancing the depressive symptoms of Asian–American midlife women through increasing physical activity.	33 midlife Asian–American immigrant women Mean age: 41.90 (9.6)	Physical activity scores and subscale scores (except active living habits scores) did not significantly change over time across groups. Depression did not significantly improve over time for both the control group and the intervention group. Intervention group experienced greater improvement in the scores over time for active living habits than control group. Both the intervention and control group had significant improvements in discrimination stress, but not in other outcome variables related to depressive symptoms.
Lesser et al. (2016) Canada	Quantitative: randomized control trial	To see if a culturally relevant dance exercise, a standard exercise and or control program could reduce the visceral adipose tissue levels.	75 postmenopausal South Asian women Mean age: 57 (6)	No significant reduction in visceral adipose tissue after 12 weeks for either the Bhangra dance intervention (−60 cm^3^, 95% confidence interval (CI) = −172–54) or standard exercise program (−98 cm^3^, 95% CI = −216 to 21). Among participants who attended more than 2/3 of exercise classes, visceral adipose tissue was significantly reduced compared to control (−109 cm^3^, 95% CI = −204 to −13, *p* = 0.026).
Im et al. (2017) USA	Quantitative: pre-test post-test	To determine the efficacy of the Web-based program in improving menopausal symptom experience of Asian–American midlife women.	29 midlife Asian–American immigrant womenMean age: 45.70 (4.0)	Total symptom severity scores decreased for control group (-0.53, *p* < 0.10), no meaningful change for the intervention group. Both study groups experienced improvement in active living habits over time, but greater among the intervention group (β = 0.29, *p* < 0.001) than the control group (β = 0.08, *p* < 0.10). Time–group interactions for total severity scores were not statistically significant anymore after controlling for physical activity (0.34 for the control group, *p* ¼ 0.10).
Lee et al. (2017) Korea	Quantitative: quasi-experimental	To investigate the long-term effectiveness of stretching exercises on the health outcomes of Korean–Chinese female migrant workers.	80 middle-aged Korean–Chinese immigrant women Mean age: 52.56 (5.45)	Effective in increasing their flexibility and decreasing work-related musculoskeletal disorder symptoms. Significant improvements in back flexibility and work-related musculoskeletal disorder symptoms after completing the intervention, but no significant differences between groups. Acculturative stress decreased at week 12 but no significant change at week 24. Enhanced intervention group maintained recommended weekly frequency of stretching until week 24 compared to the standard intervention group.
Lee et al. (2017) Korea	Quantitative: randomized control trial	To examine the effects of a standard treatment walking program compared to an enhanced treatment on cardiovascular health outcomes among middle-aged Korean–Chinese female migrant workers in Korea.	132 middle-aged Korean–Chinese immigrant women Mean age: 56.40 (5.09)	Significant decrease in 10-year risk for cardiovascular disease, blood pressure, fasting glucose, body mass index, and waist–hip ratio at 12 and 24 weeks in enhanced and standard group. Compared to baseline, walking adherence showed statistically significant increase in ST and ET groups. Mean number of steps significantly increased over time in both groups, and the change at week 24 differed between the groups. Body mass index showed greater changes in enhanced group compared to the standard treatment group.
Kim et al. (2019) Korea	Quantitative: quasi-experimental	To evaluate the effectiveness of a 24-week walking program on reducing depressive symptoms and acculturative stress levels.	132 middle-aged Korean–Chinese immigrant women Mean age: 54.60 (5.09)	There was a significant effect of walking on acculturative stress reduction for all participants. A significant interaction effect between group and time for acculturative stress was shown at weeks 12 and 24 compared to the baseline. Number of walking steps significantly increased in both the ST and the ET groups at week 12 and week 24. The participants’ depression significantly decreased at 12 and 24 weeks compared to the baseline in the ET group. A decrease in depression scores was more significant in the enhanced group at weeks 12 and 24.
Bhimla et al. (2020) USA	Quantitative: pre-test post-test	To assess the effect of a culturally relevant Zumba program on anthropometrics, physical fitness and exercise motivation among midlife Filipino women.	21 midlife Filipino–American immigrant women Mean age: 54.70 (10.829)	Significant reduction in body weight, body mass index, increase in flexibility among participants. No statistically significant differences in participants’ waist circumference, SBP, DBP, HR, and aerobic fitness. No statistically significant differences in revitalization, stress management, enjoyment, challenge, social recognition, competition, health pressures or ill-health avoidance, nimbleness, strength and endurance, appearance or positive health in the context of motivation.
Wieland (2012) USA	Mixed methods: focus groups and pre-test, post-test	To explore the effects of a community-based-participatory-research program on a sociocultural responsive fitness program to immigrant and refugee women.	45 midlife immigrant and refugee (Hispanic–, Somali– and Cambodian–American) women Mean age: 39 (N/A)	Evaluation: high acceptability of the physical activity questionnaire intervention (rated 4.85/5). Participants more likely to exercise regularly *p* ≤ 0.001), had higher health-related quality of life *p* ≤ 0.001, self-efficacy for diet (*p* = 0.36) and exercise (*p* = 0.10). Weight loss among participants, (87 vs. 83.4 kg; *p* = 0.65) decreased waist circumference (99.6 vs. 95.5 cm; *p* = 0.35), and lower blood pressure (125/80 vs. 122/76 mm/Hg; *p* = 0.27)
Vahabi and Damba (2015) Canada	Mixed methods: cohort pre-test post-test with in-depth interviews	To explore the feasibility and health impacts of implementing a culture and gender-specific physical activity programs among immigrant women in the Greater Toronto Area.	27 middle-aged South Asian–Canadian immigrant women Mean age: 42 (N/A)	Intervention posed significant physical health benefits, mental health benefits, social benefits, and culturally tailored/gender-specific benefits. Physical health status: intervention attributed to decrease in weight, waist circumference and BMI—but not statistically significant. Intervention had significant scores for physical role-functioning, vitality, and general health perception scale.
Kim et al. (2020) Korea	Mixed methods: literature review, focus group, online survey, pilot trial	To use a living lab approach for the development of mobile-based health program that focused on improving physical activity and the cultural adaptation of Korean–Chinese women workers.	16 midlife Korean–Chinese immigrant women (between 45–60 years old)	The content validity of the mobile app was found to be 0.90 and 0.96 according to the 12 KC women and 4 experts, respectively.
Lee et al. (2020)	Mixed methods	To identify how social support and social–cognitive factors are affected after a relevant lifestyle intervention delivered through social network messaging.	28 middle-aged Korean–Chinese immigrant women Mean age: 47.2 (6.5)	The perceived levels of sense of community (z = −3.30, *p* < 0.001) and social support for exercise (z = −3.09, *p* = 0.002) were significantly increased at 12 weeks compared with baseline. Qualitative findings: the most frequent type of social support through the SNS identified by this group was network support (172/259, 66.4%), followed by emotional support (40/259, 15.4%), information support (28/259, 10.8%), and esteem support (19/259, 7.3%).
Albarran (2014) USA	Qualitative: focus groups	To explore whether lifestyle behaviour interventions and promotoras facilitate behaviour change from the perspectives of the of participants.	18 midlife Hispanic–American immigrant women Mean age: 45 (8.7)	Intervention with promotoras affected three interconnected elements: self-management tools, promotora support, and new knowledge. Saw promotoras as counselors who provided emotional and social support. Intervention was emotionally therapeutic for this sample of Latinas.
Lee et al. (2015) Korea	Qualitative: semi-structured interviews—qualitative secondary data analysis	To investigate barriers to performing stretching exercise as experienced by midlife Korean–Chinese female migrant workers in Korea based on a previously conducted community-based intervention.	90 midlife Korean–Chinese immigrant women Mean age: 53.03 (5.46)	Participants experienced an average of 2.5 barriers during the study period. Intrapersonal barriers included lack of time, lack of motivation, having illness, fatigue, and lack of skill; lack of time was the barrier most commonly experienced by the participants. Work-related environmental barriers included frequent job changes, long working hours, lack of rest time, and unpredictable job demands.
Cho et al. (2017) Korea	Qualitative: intervention mapping method	To describe the progress in the development of culturally adaptive walking interventions for Korean–Chinese female migrant workers.	21 midlife Korean–Chinese immigrant women (5 selected for intervention) Mean age: N/A; middle aged*	Determinants of walking behaviour, including knowledge, self-efficacy, social support, and acculturation, were identified through an extensive literature review, community leader interviews, and a survey of female KC migrant workers. Importance of mobile phones to use for intervention. Intervention: participants made promises to themselves to exercise more after learning a typical amount of walking, and suggested encouragement text messages.
Daniel (2017) USA	Qualitative: focus groups	To examine the perspectives of midlife South Asian immigrant women related to barriers and motives for lifestyle physical activity.	40 midlife South Asian–American immigrant women Mean age: 50.6 (7.03)	Found that culturally sensitive factors such as role expectations, self-motivation need to be considered in the implementation of physical activity interventions. Motives to physical activity include self-motivation, motivational groups, optimal physical and psychological help, emphasis on external beauty and social support. Physical activity was structured and planned such as walking for exercise, riding a bike, dancing, and doing a yoga routine which supported their perceptions of physical activity as leisure time physical activity.
Choi (2020) USA	Qualitative: focus groups	To explore the barriers and facilitators associated with physical activity and to explore strategies to promote engagement in a physical activity program with an online community.	37 middle-aged Korean–American women Mean age: 48.4 (6.0)	Barriers to physical activity include individual, interpersonal, and community-level barriers. Facilitators: social influence and social support. Participants expressed benefit and interest in lifestyle modification through physical activity via the use of social media.
Cortés (2021) USA	Qualitative: description of implementation of future randomized trial	To examine the feasibility and initial efficacy of a multi-component intervention designed to reduce cardiovascular disease risk in perimenopausal Latina women.	80 perimenopausal Latina women between the ages of 40 and 55 Mean age: N/A	Trained bilingual instructors administrating intervention. Appropriate information to manage eating healthier meals and increasing daily physical activity. Physical activity incorporation: Zumba, kick boxing, walking, weights. Continued support throughout the duration of the intervention.

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
