# Peer review of "Factors Influencing Physical Activity Participation among Midlife Immigrant Women: A Systematic Review"

_ijerph, 2021, doi:10.3390/ijerph18115590_

Round 1
Reviewer 1 Report
This review summarizes the adaptation strategies of physical activity interventions for middle-aged migrant women and factors influencing them at the individual, family, and community level.
- In the Introduction, detailed descriptions of why migrant women are more vulnerable than migrant men are needed.
- Some English correction is required.
- In the Method, what age category is the middle-age life period of the inclusion criteria? ex) 35~65 years.
Reviewer 2 Report
1. Given the nice number of studies included, why a meta-analysis was
not conducted? An estimation of overall (cross-study) effects will
increase the impact.
2. In terms of conceptual implications, the authors may want to
elaborate in more detail on how current / future models are advanced in
detail.
3. The practical implications are important. The authors may want to
specify those by detailing on some concrete examples.
4. Can we expect the same findings for men? If not, why there might be
differences?
5. May findings differ by cultures / geographic region of countries of
origin / host countries?
6. Economic aspects for leaving the country of origin, and as essential
resource to be able to engage in activities in the host country etc.
seem crucial. It could be discussed in more depth.
7. Health issues and healthcare in the host country concerns a further
factor that probably deserves more elaboration in this review.
8. Perhaps it matters the marital status of the women investigated. The
social network issue could be emphasized.
Reviewer 3 Report
First of all, I would like to share the need to carry out work like the one you present. They are necessary for the advancement of science in the field they study. The purpose of the manuscript is clear and consistent. The study has been an interesting read, it is necessary to know the reality of the sector on which the work emphasizes. I find the review really interesting. I believe that the manuscript has been subject to revision before, since I have checked the previous versions of the manuscript. In the revised version in this case say that the manuscript has been substantially improved. Relevant issues have been clarified so that it strictly complies with what is required in a review work. I consider that both the work done by the authors in the improved version meets the requirements to be published.Author Response
Please see the attachment.

Reviewer 4 Report
This systematic review aims to identify influencing factors and adaption approaches of physical activity participation among midlife immigrant women. The article is too long and shallow.
English language usage is poor. Gross errors present and compromise understanding.
The introduction needs to cut in half.
Methods can stay.
The largest study shows no significant results therefore affecting the entire premise and hypothesis.
The lines and paragraphs between 213-391 can be eliminated entirely.
The discussion needs to be more concise (reduce long unnecessary sentences and cut in half)
The conclusion can remain.
Round 2
Reviewer 2 Report
I don't have further questions about the paper.
Author Response
Thank you very much for your great review.
Reviewer 4 Report
The topic is important and current. The recommended corrections were made.
Author Response

(The authors gave the same response as above.)
